# Association between Serum Zinc and Toll-like-Receptor- Related Innate Immunity and Infectious Diseases in Well-Nourished Children with a Low Prevalence of Zinc Deficiency: A Prospective Cohort Study

**DOI:** 10.3390/nu14245395

**Published:** 2022-12-19

**Authors:** Sui-Ling Liao, Man-Chin Hua, Ming-Han Tsai, Kuan-Wen Su, Chi Lin, Tsung-Chieh Yao, Li-Chen Chen, Kuo-Wei Yeh, Jing-Long Huang, Shen-Hao Lai

**Affiliations:** 1Department of Pediatrics, Chang Gung Memorial Hospital at Keelung, Keelung 204, Taiwan; 2College of Medicine, Chang Gung University, Taoyuan 333, Taiwan; 3Division of Allergy, Asthma and Rheumatology, Department of Pediatrics, Chang Gung Memorial Hospital, Taoyuan 333, Taiwan; 4Department of Pediatrics, New Taipei Municipal Tucheng Hospital, Chang Gung Memorial Hospital, Tucheng 236, Taiwan; 5Division of Pulmonology, Department of Pediatric, Chang Gung Memorial Hospital, Taoyuan 333, Taiwan

**Keywords:** serum zinc, zinc deficiency, zinc-sufficient, childhood infection, toll-like receptors, innate immunity, cytokines

## Abstract

Existing reports focus on zinc-associated immunity and infection in malnourished children; however, whether zinc also plays an important role in the immune homeostasis of the non-zinc-deficient population remained unknown. This study aimed to investigate the association between zinc status and toll-like receptor (TLR)-related innate immunity and infectious outcome in well-nourished children. A total of 961 blood samples were collected from 1 through 5 years of age. Serum zinc was analyzed, and mononuclear cells isolated to assess TNF-α, IL-6, and IL-10 production by ELISA after stimulation with TLR ligands. Childhood infections were analyzed as binary outcomes with logistic regression. The prevalence of zinc deficiency was 1.4–9.6% throughout the first 5 years. There was significant association between zinc and TLR-stimulated cytokine responses. Higher serum zinc was associated with decreased risk of ever having pneumonia (aOR: 0.94; 95% CI: 0.90, 0.99) at 3 years, and enterocolitis (aOR: 0.96; 95% CI: 0.93, 0.99) at 5 years. Serum zinc was lower in children who have had pneumonia before 3 years of age (72.6 ± 9 vs. 81.9 ± 13 μg/dL), and enterocolitis before 5 years (89.3 ± 12 vs. 95.5 ± 13 μg/dL). We emphasize the importance of maintaining optimal serum zinc in the young population.

## 1. Introduction

Over the past years in developed countries, the estimated prevalence of inadequate micronutrient intake has greatly been reduced. Thus, major clinical problems resulting from zinc deficiency might no longer be an imperative public issue. Previous reports had focused on zinc-deficiency-associated morbidity/mortality in low-income countries where the prevalence of zinc deficiency was high [1,2]. However, the role of serum zinc in the non-zinc-deficient population has rarely been explored. We hypothesized that even with proper nutrition, zinc might still play an important role in the immune homeostasis, or the prevalence of infectious diseases in young children. 

Toll-like receptors (TLRs) are highly conserved components of the innate immune system and play critical roles in early response to various invading pathogens. In vitro studies in human monocytes had shown zinc deficiency to impair TLR-4-mediated signaling that resulted in decreased cytokine responses. Studies have shown that zinc supplementation increases the secretion of IL-6, IL-10, and TNF-α. However, others have not confirmed this relationship, or have even found an inverse correlation [3,4,5]. Clinically, zinc deficiency was associated with an increased risk of childhood infections in lower-income countries; and supplementing with zinc had shown promising results in preventing or treating infections such as diarrhea and pneumonia [1,6,7]. Recent reports have also suggested a potential therapeutic effect of zinc in patients with COVID-19 infection due to its role in the innate immunity [8,9]. Early childhood is characterized by increased susceptibility to various invading pathogens due to the relatively immature immune system. Thus, maintaining a higher serum level might be potentially effective in preventing serious infections in the young population. 

In the current study, we aimed to investigate the association between zinc status and common childhood infections in a prospective cohort study conducted in a developed country where zinc deficiency was uncommon. In addition, existing reports on zinc-associated innate immunity other than TLR4-related signaling pathway are limited [3,4,10]. Thus, we also aimed to investigate the relationship between serum zinc and several TLR-triggered cytokine responses throughout early childhood, the period during which nutrition plays an important role in the developing immune system.

## 2. Materials and Methods

### 2.1. Study Population

This study is part of an ongoing prospective birth cohort study called the PATCH (The Prediction of Allergy in Taiwanese Children), which enrolled participants residing in the cities of northern Taiwan. It is an unselected, population-based study designed to investigate early protective measures to prevent the development of infectious or allergic diseases later in life. The study was approved by the Chang Gung Ethics Committee, and informed consent was obtained from the parents/legal guardians of the neonates. Starting from March 2015, pregnant women were invited randomly by a study nurse during their third-trimester visits to join our research program. Children were followed as scheduled since birth. However, many children missed their scheduled visit during the past 2 years due to the COVID-19 pandemic, and many clinical data/blood samples for the specific age group were not available. Thus, for the current analysis, only participants with available blood tests at each age period were included in the cross-sectional analysis and further analyzed with a generalized estimating equation. The detailed number of participant enrollment and blood specimens is listed in Figure 1. 

Questionnaires with regard to infectious diseases were given at 1, 2, 3, and 5 years of age. Infectious outcomes such as pneumonia, croup, acute otitis media, infectious enterocolitis, and urinary tract infection were defined as ever having an infection since birth if there was a diagnosis from a doctor and had either been hospitalized or received medical treatment for the disease. For the cross-sectional analysis, the incidence of ever having an infection at 1 year of age was defined as diseases occurring between 0 and 1 y; at 3 years, the incidence between 0 and 3 y; and at 5 years, the incidence between 0 and 5 y. Medical charts or records from the National Health Insurance database were retrieved to confirm the diagnosis in those who reported an infection. The definition of pneumonia was based on the primary physicians’ diagnosis, which generally included clinical symptoms and radiographic findings compatible with pneumonia. Definitions of infectious enterocolitis and urinary tract infections were based on positive proof of an infectious organism, or if the patient had been hospitalized due to disease severity.

### 2.2. Blood Collection and Zinc Measurement

Blood samples were collected yearly during the first 5 years to determine the serum zinc concentration and every other year to harvest mononuclear cells for TLR ligand stimulation. Analysis of serum zinc was performed in the central clinical laboratory of our hospital, which was performed by atomic absorption (AA) spectroscopy by PerkinElmer PinAAcle 900T (PerkinElmer SCIEX, Waltham, MA, USA). Zinc standards were purchased from HIGH-PURITY STANDARDS (North Charleston, SC, USA) at a concentration of 1000 ppm. Glycerol was purchased from Sigma-Aldrich (St. Louis, MO, USA), with a purity of ACS reagent ≥ 99.5%. To avoid the effect of infection on measured values, drawings of blood samples were delayed if children had any signs of infection (including fever, rhinorrhea, cough, diarrhea, or any other discomforts suspicious of infection) within 3 weeks of blood testing. 

### 2.3. Toll-like Receptor Ligands Stimulation 

Details of mononuclear cells isolation were described previously with Ficoll-Hypaque (Pharmacia Biotech, Piscataway, NJ, USA) [11]. TLR ligands used for cell stimulation were obtained from InvivoGen (San Diego, CA, USA), which included synthetic bacterial lipoprotein (PAM3csk4) that was recognized by TLR1–2; a synthetic analog of double-stranded RNA for TLR3 (poly I:C); ultrapure LPS for TLR4; and R848 that activated via the TLR7/TLR8 signaling pathway. The concentrations of the ligands used for this experiment were as follows: 10 µg/mL of PAM3csk4, 10 µg/mL of poly (I:C), 20 µg/mL of LPS, and 10 µg/mL of R848. Cells were also treated with the NF-kB activator phytohemagglutinin (Murex Pharmaceuticals, Dallas, TX, USA) at 4 µg/mL in R10-FBS as a positive control.

### 2.4. Measurement of TLR-Stimulated Cytokines 

To determine TLR responses, 3 × 10^5^ PBMCs in 100 µL of R10-FBS (or R10-HS, or R10 without serum, where specified) were added to each of the duplicate ligand- or medium-containing wells and incubated at 37 °C with 5% CO_2_. All assay preparations were performed using a sterile technique in a laminar flowhood. Cultured supernatants were harvested after 20 h of incubation, and levels of TNF-α, IL-6, and IL-10 were determined by enzyme-linked immunosorbent assays (ELISA; R&D systems, MN; Cat. STA00D, S6050, S1000B, respectively). The detection limits were 15.6 pg/mL for TNF-α, 3.12 pg/mL for IL-6, and 7.8 pg/mL for IL-10.

### 2.5. Outcome Assessment 

The primary outcomes of the study were defined as ever having an infection before each age period (pneumonia, croup, acute otitis media, enterocolitis, and urinary tract infection). Secondary outcomes were the effect of serum zinc concentration on TLR-triggered cytokine profiles throughout the first five years. In addition, since zinc is an important nutrient for childhood growth, the incidence of failure to thrive and stunted growth was also investigated.

### 2.6. Statistical Methods

Logistic regression models were used to determine the association between serum zinc at each age period and binary outcomes (defined as ever having at least one episode of pneumonia, croup, acute otitis media, infectious enterocolitis, and/or urinary tract infection). Serum zinc levels in children with the disease (pneumonia or enterocolitis) and those without were further compared by using student’s *t*-test. The associations between serum zinc and TLR-stimulated cytokine level were analyzed by using linear regression models. As cytokine levels were not normally distributed, values were logarithmically transformed as continuous variables in the statistical models. To compensate for confounders’ effects, characteristics such as gestational age, gender, birth body weight, current growth parameter, mode of delivery, maternal education, day-care attendance, and breastfeeding duration were included in the analyses. All statistical analyses were carried out by using IBM SPSS Statistics version 25 (Armonk, NY, USA). A 2-sided *p* value < 0.05 was used to determine statistical significance.

## 3. Results

### 3.1. Study Population Characteristics 

Due to a substantial amount of missing data, only participants with both clinical data and blood tests were included in the current analysis. Blood samples were collected starting at the age of 1 year for 5 years (1 y, *n* = 227; 2 y, *n* = 241; 3 y *n* = 245; 5 y, *n* = 234). Because the cytokine study was not performed in the 2-year-old children, thus, they were further excluded in the outcome analysis. The incidence of ever having various common childhood infections throughout the first 5 years of life is listed in Table 1. Our participants were mostly well nourished and compatible with the low prevalence of zinc deficiency; very few children had weight or height-for-age falling below the fifth percentile of the WHO growth curve [12]. 

### 3.2. Serum Zinc Status throughout Early Childhood

The prevalence of zinc deficiency was 1.3–9.6% throughout the first 5 years (defined as serum zinc < 65 μg/dL for children <10 years of age [12]. Although median zinc level appeared to increase with age, however, compared to other age groups, the occurrence of having low serum zinc levels (<65 μg/dL) was higher at the age of 3 years (9.6%), and reduced to only 1.4% by age 5 years. (Table 2). 

### 3.3. Effect of Serum Zinc Status on TLR-Triggered Cytokine Production throughout Early Childhood

Linear regression analysis was used to determine the association between serum zinc concentration and TLR-triggered TNF-α, IL-6, and IL-10 response in children throughout the first 5 years. Serum zinc status was not associated with any of the TLR-ligand-stimulated cytokine responses in the 1-year-old infants, as shown in (Appendix A. However, by 3 years of age, there was a significant association between serum zinc concentration and TLR1/2-, 4- and 7/8-stimulated IL-6 responses (*p* < 0.01 for all ligands), and TLR1/2- and 7/8-stimulated IL-10 responses (*p* = 0.01 for all ligands). Serum zinc was also associated with cytokine responses in the 5-year-old children, but only with TLR1/2-, 4- and 7/8-triggered TNF-α responses (*p* = 0.01 for TLR 1/2, *p* < 0.01 for TLR4, and *p* = 0.01 for TLR7/8 ligands) (Table 3). 

### 3.4. Association between Serum ZINC Status and Childhood Infections

The association between serum zinc and infectious diseases was investigated throughout the first five years with logistic regression analysis. Corresponding to the null results of the cytokine profile at the age of 1 year, serum zinc concentration was not associated with the prevalence of infectious diseases during infancy. However, by the age of 3 years, higher serum zinc was correlated with decreased risk of ever having pneumonia (aOR: 0.94; 95% CI: 0.90, 0.99; *p* = 0.01). An association was noted between serum zinc and the prevalence of urinary tract infections in the 3-year-old children (OR: 0.96; 95% CI: 0.92, 0.99; *p* = 0.01), but was not statistically significant after adjusting for confounders (aOR: 0.97; 95% CI: 0.93, 1.01; *p* = 0.09). Serum zinc was also associated with a decreased risk of ever having infectious enterocolitis in the 5-year-old children (aOR: 0.96; 95% CI: 0.93, 0.99; *p* = 0.02) (Table 4). For longitudinal analysis, a generalized estimating equation (GEE) was used to estimate the association between serial zinc level and disease outcome throughout the first 5 years. Results had also shown zinc status to be associated with the prevalence of pneumonia and enterocolitis during early childhood (Appendix A).

### 3.5. Comparing Serum Zinc Status in Children with and without Infectious Diseases

The serum zinc level was compared between children with or without ever having pneumonia and infectious enterocolitis in Figure 2. The result showed that children who have had pneumonia during the first 3 years had a lower mean serum zinc concentration by the age of 3 years. Their mean serum zinc serum concentration was 72.6 ± 9 μg/dL compared to 81.9 ± 13 μg/dL in those children without pneumonia. The mean serum zinc serum was 89.3 ± 12 μg/dL in the 5-year-old children who have had infectious enterocolitis compared to 95.5 ± 13 μg/dL in those without severe diarrhea. 

## 4. Discussion

The results from this study showed that serum zinc status was associated with altered cytokine responses to various toll-like receptor ligands (TLR1/2, 4, 7/8) and infectious diseases (pneumonia and enterocolitis) in a prospective cohort of children who were mostly zinc-sufficient. Previous reports have shown zinc deficiency to be associated with an increased risk and severity of diarrhea and pneumonia in low-income countries due to malnutrition [1,14,15,16]. However, whether zinc status still has an impact on the immune function and clinical outcome in children of the developed countries where nutritional sources are abundant is unknown. This study was among the few to explore the role of zinc in a population where the prevalence of zinc deficiency was low. In addition, this is the first study to investigate the relationship between serum zinc concentration and several TLR-triggered cytokine responses. Our result had shown lower serum zinc to be associated with reduced cytokine responses to cellular stimulation with TLR1/2, TLR4, and TLR7/8 ligands. Such results pointed to a possibility that suboptimal serum zinc status might reduce cytokine responses to various invading pathogens, thus resulting in an increased risk of infections. Our result was supported by studies that had shown zinc supplementation to increase the secretion of IL-6, IL-10, and TNF-α in a dose-dependent response [5].

The important role of zinc in the innate immunity has been widely explored [16,17,18,19,20]. Similar to our results, studies in both murine and human primary monocytes had shown zinc deficiency to impair TLR4-mediated signaling that resulted in reduced production of cytokines such as TNF-α, IL-6, IL-10, and Il-1β. However, despite extensive studies on the role of zinc as a signaling molecule in the toll-like receptor pathway, most studies focused only on TLR4-related signaling (LPS) [3,17,18,21]. To expand the knowledge of previous studies, we had investigated the role of zinc in several other TLR pathways. Our results showed zinc to be also involved in the TLR1/2, and TLR7/8 signaling. These two pathways play key factors in acting as a first line of defense against triacyl lipopeptides and viral nucleic acids. Recent studies have shown SARS-CoV-2 to induce pro-inflammatory cytokines through the TLR2, TLR4, and TLR7/8 signaling [22,23,24,25]. These reports supported the potential therapeutic role of zinc against COVID-19 infection, emphasizing the important role of zinc in the TLR-related immunity.

In regard to common childhood infections, similar to previous reports, we had found higher serum zinc to be associated with a decreased risk of pneumonia and enterocolitis [2,6,14]. Clinical studies had shown that supplementing zinc in malnourished children had successfully reduced the duration and persistence of severe diarrhea and the incidence of acute lower respiratory tract infections [26,27,28]. In one randomized control study from India, using zinc as adjunct treatment had reduced 40% treatment failure in children with severe bacterial illness [29]. Nevertheless, most RCTs were conducted in malnourished children from the low- or middle-income countries; thus, whether routine zinc supplementation has any beneficial effects in children with adequate nutritional provision remained unknown. Even though our result showed higher zinc status to protect against severe infections, because supplementing zinc in high doses might have negative effects on the immune system (such as suppression of T cell function or blocking IFN- α production [29], further randomized studies are thus needed to determine the proper supplemental dosage or optimal serum level for infection prevention in well-nourished children.

The strength of the current manuscript is that we are among the first ones to provide laboratory and clinical evidence of the role of serum zinc in a cohort of young children without malnutrition. However, there are some limitations. First, because serum zinc concentration might be affected by recent meals and tissue catabolism, it might not always be a reliable indicator of an individual’s true zinc status [30]. Thus, it is possible that we might have had misallocated some children as having sufficient zinc due to a recent zinc-abundant meal. Nevertheless, a systemic review had concluded that serum zinc level was responsive to both supplement and depletion, and thus remained the most widely used biomarker [31]. Secondly, our study design was unable to assign causation, and the relatively small number of children having infections might have limited the power to assess differences in any disease outcomes related to differences in zinc status, thus raising a possibility that some associations might arise by chance. Finally, our study on the association between zinc and the whole TLR-signaling pathway was incomplete (lacking TLR5–6 and 9–10), and the role of zinc in the adaptive immunity was not investigated. Further studies are needed to better understand the role of zinc in both the innate and adaptive immune pathway.

## 5. Conclusions

In conclusion, our results pointed out that, despite having an adequate serum level, zinc status was still associated with an altered innate cytokine response and infectious outcome in well-nourished children. Thus, efforts should be given to ensure proper amount of zinc-rich food in children’s daily dietary regimen. Currently, we cannot recommend the use of zinc supplementation, yet we suggest further interventional trials to determine whether zinc supplementation can improve immunity in this population. In addition, future studies are also needed to determine the optimal serum zinc level for infection prevention in young children of the developed countries.

## Figures and Tables

**Figure 1 nutrients-14-05395-f001:**
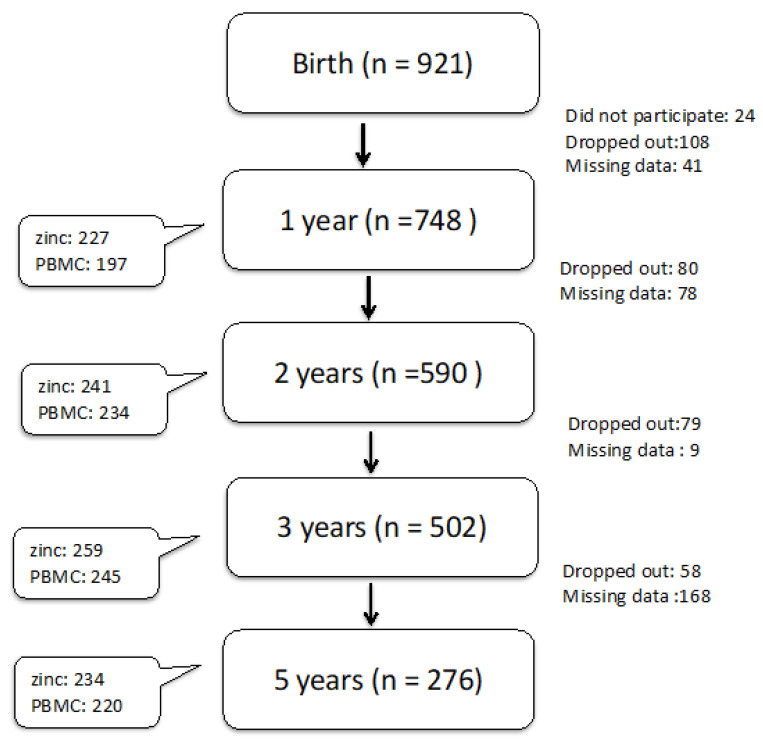
Flowchart of the cohort study: valid questionnaire information and medical records were complete for all children at various ages in the main box. The side box demonstrates the number of blood samples with 1 sample/child at each specific age period for the analysis of zinc and PBMC (peripheral mononuclear cell) isolation for the cytokine stimulation tests. Missing data: mostly because participants missed the clinical visit at that specific age period or clinical data (medical records and/or anthropometric measurements) were incomplete.

**Figure 2 nutrients-14-05395-f002:**
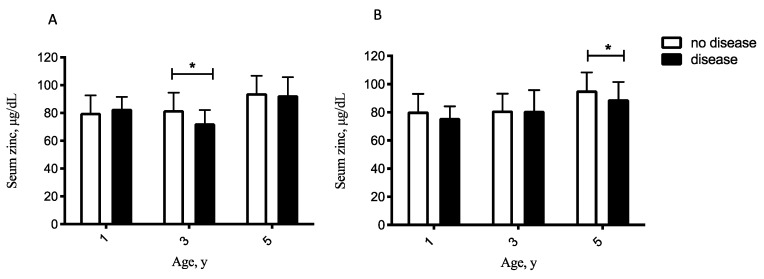
Comparing serum zinc concentration between children who have had pneumonia (**A**) and infectious enterocolitis (**B**) before the ages of 1, 3, and 5 years with those having never suffered from the above diseases. Values represent means ± SD. * means differ at that age, *p* < 0.05.

**Table 1 nutrients-14-05395-t001:** Demographic data of the study population.

Characteristic	Age 1 y (*n* = 227)	Age 2 y (*n* = 241)	Age 3 y (*n* = 259)	Age 5 y (*n* = 234)
Male	147 (64.8)	145 (60.1)	158 (61.0)	149 (63.7)
BF > 12 mo	44 (19.3)	34 (14.1)	34 (13.1)	45 (19.2)
Body weight (Kg)	9.7 ± 1	12.8 ± 2	14.9 ± 2	19.4 ± 3
Body height (cm)	76 ± 3	90 ± 4	97 ± 5	111 ± 5
Failure to thrive *n* (%)	2 (1.6)	4 (1.7)	5 (1.9)	6 (2.6)
Stunted growth *n* (%)	6 (2.6)	6 (2.5)	11 (4.2)	3 (1.3)
Infectious diseases				
Pneumonia *n* (%)	7 (3.1)	17 (7.1)	28 (10.8)	37 (15.9)
Croup *n* (%)	4 (1.8)	7 (2.9)	14 (5.4)	17 (7.2)
AOM *n* (%)	6 (2.6)	16 (6.6)	26 (10.0)	52 (22.2)
Enterocolitis *n* (%)	13 (5.7)	25 (10.4)	39 (15.1)	53 (22.6)
UTI *n* (%)	16 (7.0)	24 (9.6)	29 (11.2)	33 (14.1)

^1^ BF > 12 mo: breastfeeding duration for more than 12 months. ^2^ Failure to thrive: defined as weight-for-age falling below the 5th percentile of the WHO growth chart [13]. ^3^ Stunted growth: defined as height-for-age falling below the 5th percentile of the WHO growth chart [13]. Abbreviations: BF, breastfeeding; AOM, acute otitis media; UTI, urinary tract infection.

**Table 2 nutrients-14-05395-t002:** Serum zinc status throughout the first 5 years.

Age, y	Samples *n*	Median (Range)	Zinc < 65 μg/dL *n* (%)	Zinc > 120 μg/dL *n* (%)
1	227	79.3 (55.7–158)	18 (7.9)	1 (0.4)
2	241	80.3 (50.3–112)	16 (6.6)	0
3	259	80.3 (48.5–121)	25 (9.6)	1 (0.4)
5	234	95.3 (40.2–128)	3 (1.4)	6 (2.8)
Total	961	84.0 (40.2–158)	62 (6.5)	8 (0.8)

**Table 3 nutrients-14-05395-t003:** Association between serum zinc concentration and TLR-stimulated cytokine responses in children at ages 3 and 5 years.

	Univariate Analysis				Multivariate Analysis			
	3 y (95% CI)	*p*	5 y (95% CI)	*p*	3 y (95% CI)	*p*	5 y (95% CI)	*p*
*TLR1–2*								
TNF-α	0.002 (−0.01, 0.02)	0.82	−0.02 (−0.04, −0.01)	<0.01	0.01 (−0.01, 0.03)	0.23	−0.02 (−0.04, −0.01)	0.01
IL-6	−0.01 (−0.02, −0.004)	<0.01	−0.001 (−0.01, 0.01)	0.82	−0.02 (−0.03, −0.01)	<0.01	−0.002 (−0.01, 0.01)	0.68
IL-10	−0.02 (−0.03, 0.00)	0.05	−0.003 (−0.01, 0.01)	0.53	−0.03 (−0.05, −0.01)	0.01	−0.001 (−0.01, 0.01)	0.91
*TLR3*								
TNF-α	0.02 (−0.01, 0.05)	0.18	−0.002 (−0.03, 0.03)	0.89	0.02 (−0.02, 0.06)	0.37	−0.003 (−0.04, 0.03)	0.84
IL-6	−0.01 (−0.03, 0.02)	0.61	0.01 (−0.01, 0.02)	0.52	−0.01 (−0.05, 0.02)	0.49	0.01 (−0.01, 0.03)	0.49
IL-10	0.01 (−0.01, 0.03)	0.47	−0.01 (−0.02, 0.01)	0.51	−0.01 (−0.03, 0.02)	0.68	−0.01 (−0.03, 0.01)	0.42
*TLR4*								
TNF-α	0.002 (−0.01, 0.01)	0.65	−0.01 (−0.01, −0.002)	0.01	0.002 (−0.01, 0.01)	0.65	−0.01 (−0.02, −0.004)	<0.01
IL-6	−0.01 (−0.02, −0.01)	<0.01	0.00 (−0.002, 0.002)	0.74	−0.01 (−0.02, −0.01)	<0.01	−0.001 (−0.003, 0.002)	0.52
IL-10	−0.01 (−0.02, 0.006)	0.29	−0.01 (−0.02, 0.003)	0.17	−0.01 (−0.03, 0.006)	0.18	−0.005 (−0.02, 0.01)	0.38
*TLR7–8*								
TNF-α	0.01 (−0.01, 0.004)	0.30	−0.01 (−0.02, −0.001)	0.03	0.007 (−0.02, 0.01)	0.27	−0.01 (−0.02, −0.003)	0.01
IL-6	−0.01 (−0.01, −0.004)	<0.01	−0.001 (−0.01, 0.003)	0.71	−0.01 (−0.02, −0.006)	<0.01	−0.002 (−0.01, 0.003)	0.46
IL-10	−0.02 (−0.03, −0.001)	0.04	−0.004 (−0.02, 0.01)	0.48	−0.03 (−0.05, −0.01)	0.01	−0.004 (−0.02, 0.01)	0.58

^1^ Adjusted for gestational age, gender, birth body weight, mode of delivery, maternal allergy, age of solid food introduction, and breastmilk duration. Abbrev: IL: interleukin; TLR: toll-like receptor; TNF: tumor necrosis factor; y: years. ^2^ Only participants with both zinc and cytokine data at the specific age period were included in the analysis: (1 y, *n* = 197; 3 y, *n* = 234; 5 y, *n* = 220).

**Table 4 nutrients-14-05395-t004:** Association between serum zinc concentration and clinical infectious outcome throughout the first 5 years.

	Crude OR (95% CI)	*p*	Adjusted OR (95% CI)	*p*
1 y				
Pneumonia	1.02 (0.96–1.07)	0.60	1.09 (0.95–1.2)	0.24
Croup	1.05 (0.99–1.10)	0.08	1.10 (0.97–1.24)	0.12
AOM	1.01 (0.94–1.08)	0.84	1.06 (0.96–1.18)	0.26
Enterocolitis	0.96 (0.91–1.02)	0.22	0.97 (0.91–1.03)	0.33
UTI	0.95 (0.90–1.00)	0.07	0.94 (0.88–1.00)	0.05
3 y				
Pneumonia	0.94 (0.90–0.98)	0.001	0.94 (0.90–0.99)	0.01
Croup	0.97 (0.93–1.01)	0.17	0.97 (0.91–1.03)	0.29
AOM	0.99 (0.96–1.03)	0.66	0.99 (0.96–1.03)	0.73
Enterocolitis	1.00 (0.98–1.03)	0.99	1.00 (0.95–1.04)	0.83
UTI	0.96 (0.92–0.99)	0.01	0.97 (0.93–1.01)	0.09
5 y				
Pneumonia	0.99 (0.96–1.02)	0.53	1.00 (0.95–1.03)	0.59
Croup	0.96 (0.93–1.00)	0.07	0.98 (0.92–1.03)	0.38
AOM	1.01 (0.98–1.04)	0.60	1.01 (0.98–1.04)	0.68
Enterocolitis	0.96 (0.94–0.99)	0.01	0.96 (0.93–0.99)	0.02
UTI	1.03 (1.00–1.07)	0.06	1.04 (1.00–1.09)	0.06

Adjusted for gestational age, gender, current body weight, mode of delivery, maternal education, day-care attendance, and breastmilk duration. AOM: acute otitis media; UTI: urinary tract infection.

## Data Availability

The data will not be openly available because they contain personal information of the participants. However, the analytical code used for the current study can be made available upon reasonable request and the data are open for potential research collaboration by contacting the corresponding author.

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
