# Peer review of "Association between Serum Zinc and Toll-like-Receptor- Related Innate Immunity and Infectious Diseases in Well-Nourished Children with a Low Prevalence of Zinc Deficiency: A Prospective Cohort Study"

_nutrients, 2022, doi:10.3390/nu14245395_

Round 1

Reviewer 1 Report

The manuscript deals with an interesting topic and is quite interesting itself. However, the manuscript is not well-written as there are many grammatical mistakes, disjointed statements and lack of orderliness. The following issues need to be addressed.

1. Many studies have linked the serum concentration of Zn with the concentration of certain cytokines. However, some studies have shown that zinc supplementation increases the secretion of IL-6, IL-10, and TNF-α, while others have not confirmed this relationship, or have even found an inverse correlation. This issue should be developed in the introduction and in the discussion.

2. Serum zinc concentration was not correlated with the children's diet. Were children supplemented with mineral supplements (containing zinc)? Was such information collected?

3. Abstract is too long (up to 317 words) - inconsistent with the journal's guidelines.

4. Line 48 - Latin names should be written in italics (in vitro).

5. Figure 1 - zinc or Zinc - not interchangeable.

6. Wrong way of citing – brackets should be instead of parentheses.

7. In section 2.2, the sample preparation and the test conditions by atomic absorption spectroscopy should be described.

8. Item 2.4 - provide details of the ELISA tests used (including catalog numbers - line 127)

9. All abbreviations used in tables and figures should be explained below (Table 3, 4…).

10. Line 207 - table 4 (bold).

11. Sentences always start with a capital letter (line 220).

12. References section - formatting inconsistent with the journal's guidelines.

13. Authors should reduce generic discussion and focus more on interpreting their data while comparing it with literature.

Author Response

Dear Reviewer 1:

Thank you so much for your wise and detailed correction of the manuscript. We had revised the text by following your recommendations and hope we had reached your high standards. The following are answers to your questions:

  1. Many studies have linked the serum concentration of Zn with the concentration of certain cytokines. However, studies have shown that zinc supplementation increases the secretion of IL-6, IL-10, and TNF-α, while others have not confirmed this relationship, or have even found an inverse correlation. This issue should be developed in the introduction and in the discussion

Ans.: Statement was added to the section of introduction.

  1. Serum zinc concentration was not correlated with the children's diet. Were children supplemented with mineral supplements (containing zinc)? Was such information collected?

Ans.: Unfortunately, details on children’s diet were not collected

  1. Abstract is too long (up to 317 words) - inconsistent with the journal's guidelines.

Ans.: The abstract is currently 298 words.

  1. Line 48 - Latin names should be written in italics (in vitro).

Ans.: Corrections made

  1. Figure 1 - zinc or Zinc - not interchangeable.

Ans.: Corrections made

  1. Wrong way of citing – brackets should be instead of parentheses.

Ans.: Corrections made

  1. In section 2.2, the sample preparation and the test conditions by atomic absorption spectroscopy should be described.

Ans.: Description was given in the methods

  1. Item 2.4 - provide details of the ELISA tests used (including catalog numbers - line 127)

Ans.: Catalog numbers given in the methods

  1. All abbreviations used in tables and figures should be explained below (Table 3, 4…).

Ans.: Abbreviations given

  1. Line 207 - table 4 (bold)

Ans.: Sorry, we did not understand your request

  1. Sentences always start with a capital letter (line 220).

Ans.: Corrections made

  1. References section - formatting inconsistent with the journal's guidelines.

Ans.: Corrections made

  1. Authors should reduce generic discussion and focus more on interpreting their data while comparing it with literature.

Ans.: Corrections made

Reviewer 2 Report

Manuscript nutrients-2003031 is a cohort study that reports on the importance of mainting zinc levels in children population without malnutrition in order to avoid infections and illnesses like pneumonia or any other health disorder.

The manuscript is relevant to the aims and scope of Nutrients MDPI journal and is in generally well structured and designed. The authors have discussed the importance and the limitations of their study, which in my opinion is very important for the readers. The work is novel in nature and may comprise the basis for future research regarding the levels of zinc serum in young children and its role in diseases prevention. Eventhough, the manuscript is of high quality it requires a revision.

Firstly, the authors must improve the use of English language as there are grammar errors. This is importnat to avoid the loss of citations. In addition, there must be a statement regarding the Ethics Committee of their University for the agreement to carry out this study as it involves human participants.

I have indicated within the attached pdf the corrections that should be done. However, the authors must proof-read line by line the whole manuscript for grammar errors.

Based on the above, I suggest a minor revision prior to further consideration for publication.

Author Response

Dear Reviewer 2:

Thank you so much for your wise and detailed correction of the manuscript. We had revised the text by following your recommendations and hope we had reached your high standards. A statement regarding the Ethics Committee was given in the text. Thank you so much for your valuable time and corrections.

Round 2

Reviewer 1 Report

1. In the introduction, only a sentence from the reviewer's statement was inserted. However, the reviewer asked for the issue of zinc supplementation on the secretion of IL-6, IL-10 and TNF-α to be developed, both in the introduction and in the discussion. This is still missing!

2. The authors state that they have no information on the children's diet (including supplementation with preparations containing zinc). If the serum zinc concentration was not correlated with the children's diet, it is difficult to draw correct conclusions. I consider this to be a serious shortcoming and therefore cannot accept such a study.

3. The discussion was not improved as suggested by the reviewer. Even the abstract was not revised in accordance with the journal's guidelines.

Author Response

Dear reviewer; Thank you so much for your wise and detailed correction of the manuscript. We had revised the text by following your recommendations and hope we had reached your high standards. The following are answers to your questions:

  1. In the introduction, only a sentence from the reviewer's statement was inserted. However, the reviewer asked for the issue of zinc supplementation on the secretion of IL-6, IL-10 and TNF-α to be developed, both in the introduction and in the discussion. This is still missing

Ans.: Reference was added to the introduction and discussion

  1. The authors state that they have no information on the children's diet (including supplementation with preparations containing zinc). If the serum zinc concentration was not correlated with the children's diet, it is difficult to draw correct conclusions. I consider this to be a serious shortcoming and therefore cannot accept such a study.

Ans.: Our study focused on well-nourished children, thus, we had very few children with zinc deficiency and failure to thrive. Thus, we did not realize diet could affect the study conclusion, In addition, the study consists of children from 1 year to 5 years of age, thus, diet varied greatly among the children, and comparisons could not be made

  1. The discussion was not improved as suggested by the reviewer. Even the abstract was not revised in accordance with the journal's guidelines.

Ans.: Abstract was revised according to the guideline